# The Ribosome Is the Ultimate Receptor for Trypsin Modulating Oostatic Factor (TMOF)

**DOI:** 10.3390/biom12040577

**Published:** 2022-04-14

**Authors:** Dov Borovsky, Pierre Rougé, Robert G. Shatters

**Affiliations:** 1Department of Biochemistry and Molecular Genetics, School of Medicine, University of Colorado Anschutz, Aurora, CO 80045, USA; 2UMR 152 Pharma-Dev, Faculté des Sciences Pharmaceutiques, Université Toulouse 3, CEDEX 09, F-31062 Toulouse, France; pierre.rouge@free.fr; 3USDA ARS, U.S. Horticultural Research Laboratory, Subtropical Insects and Horticulture Research Unit, 2001 Rock Road, Fort Pierce, FL 34945, USA; robert.shatters@usda.gov

**Keywords:** mosquito, ribosome binding, *Aea*TMOF, oncocin112 (1–13), molecular modeling, kinetic characterization, in vitro translation and inhibition

## Abstract

*Aedes aegypti* Trypsin Modulating Oostatic Factor (*Aea*TMOF). a mosquito decapeptide that controls trypsin biosynthesis in female and larval mosquitoes. enters the gut epithelial cells of female mosquitoes using ABC-*tmf*A receptor/importer. To study the ultimate targeted receptor after *Aea*TMOF enters the cell, *Aea*TMOF was incubated in vitro with either *Escherichia coli* or *Spodoptera frugiperda* protein-expressing extracts containing 70S and 80S ribosomes, respectively. The effect of *Aea*TMOF on luciferase biosynthesis in vitro using 70S ribosomes was compared with that of oncocin112 (1–13), a ribosome-binding antibacterial peptide. The IC_50_ of 1 μM and 2 μM, respectively, for both peptides was determined. Incubation with a protein-expressing system and *S. frugiperda* 80S ribosomes determined an IC_50_ of 1.8 μM for *Aedes aegypti* larval late trypsin biosynthesis. Incubation of purified *E. coli* ribosome with increasing concentration of *Aea*TMOF shows that the binding of *Aea*TMOF to the bacterial ribosome exhibits a high affinity (K_D_ = 23 ± 3.4 nM, B_max_ = 0.553 ± 0.023 pmol/μg ribosome and K_assoc_ = 4.3 × 10^7^ M^−1^). Molecular modeling and docking experiments show that *Aea*TMOF binds bacterial and *Drosophila* ribosome (50S and 60S, respectively) at the entrance of the ribosome exit tunnel, blocking the tRNA entrance and preventing protein biosynthesis. Recombinant *E. coli* cells that express only ABC-*tmf*A importer are inhibited by *Aea*TMOF but not by oncocin112 (1–13). These results suggest that the ribosome is the ultimate targeted receptor of *Aea*TMOF.

## 1. Introduction

*Aedes aegypti* Trypsin Modulating Oostatic Factor (*Aea*TMOF), a proline-rich peptide (YDPAPPPPPP), was originally isolated from the ovaries of female *Ae. aegypti*. *Aea*TMOF controls the biosynthesis of trypsin-like enzymes in the midgut epithelial cells of several species of mosquito including *Aedes. aegypti, Culex quinquefasciatus, Culex. nigripalpus,* and *Anopheles albimanus* [1]. *Aea*TMOF circulates in the hemolymph of female *Ae. aegypti* and *Cx. quinquefasciatus* after the blood meal and affects the translation of the late trypsin transcript in the gut epithelial cells [2,3]. *Aea*TMOF was also shown to inhibit the translation of the late trypsin transcript in *Neobellieria* [4] and in *Heliothis virenscens*. This was observed by expressing *Aea*TMOF on the coat protein of Tobacco Mosaic Virus and feeding it to the insects, where it stopped the translation of the trypsin transcript in the gut epithelial cells [5]. These earlier results indicated that *Aea*TMOF affects the translation of the trypsin transcript but does not affect the transcript abundance by inducing mRNA decay [6]. Therefore, it is possible to suggest that the effect is exerted on the ribosome where the translation of the trypsin message is blocked. Indeed, our earlier results show that *Aea*TMOF secreted by the mosquito ovaries binds to a *Aea*TMOF ABC gut receptor (ABC*tmf*A receptor/importer) that imports *Aea*TMOF into the gut epithelial cells [7]. *Aea*TMOF is a proline-rich peptide analogous to the antimicrobial proline-rich peptides (PR-AMP) that were first reported in honeybees [8] and subsequently identified in species including mammals, amphibians, crustaceans, and mollusks [9,10,11,12]. Initial efforts to find the bacterial target for PR-AMP identified a heat shock protein DnaK as the main candidate for inhibition; however, subsequent studies showed that *dnaK* deleted mutants were as efficiently inhibited by oncocin (Onc72 and Onc112) and apidaecin (Api88 and Api137) derivatives as the parental strain [13]. Further studies by these authors indicate that these PR-AMP target the bacterial cell ribosome. Several studies using X-ray crystallography showed that the first 13 amino acids of Onc112 bind within the ribosomal exit tunnel extending into the peptidyl transferase center overlapping with the binding site of the aminoacyl-tRNA, and thus preventing entry into the elongation phase. [14,15]. To find out the mechanism by which *Aea*TMOF stops the translation of trypsin in the female *Ae. aegypti* midgut epithelial cells, we used molecular modeling of *Escherichia coli* and *Drosophila melanogaster* ribosome with *Aea*TMOF to find out whether *Aea*TMOF-like oncocin binds the peptide exit tunnel as was shown by X-ray crystallography for oncocin binding to the 70S ribosomes of *Thermus thermophilus* [14,15]. We combined molecular modeling with binding studies of *Aea*TMOF to purified *E. coli* ribosomes in combination with in vitro studies using *Spodoptera frugiperda* and *E. coli* cell free translation of *Ae. aegypti* larval late trypsin and luciferase to find out how *Aea*TMOF stops trypsin and luciferase translation in vitro. We compared these results by using a short oncocin112 peptide with the first 13 amino acids, naming it oncocin112 (1–13) (VDKPPYLPRPRPP). These amino acids were shown by X-ray diffraction to occupy the peptide exit tunnel of the *T. thermophilus* ribosome [14,15]. The *E. coli-*purified ribosome was incubated with *Aea*TMOF-FITC, and the K_D_ of *Aea*TMOF binding to the ribosome was determined showing that the binding affinity of *Aea*TMOF to the ribosome is high. This report shows for the first time the mechanism that *Aea*TMOF employs to modulate trypsin biosynthesis in the mosquito epithelial cells.

## 2. Materials and Methods

### 2.1. Bacterial Strains, and Chemicals

*E. coli* CGSC strain7636: F^−^ Δ(araD-araB)567,ΔlacZ4787(::rrnB-3),l^−^,rph-1, *Δ(rhaD-rhaB)568, hsdR514* and *E. coli* CGSC strain 8547: F^−^
*Δ(araD-araB)567, ΔlacZ4787*(::rrnB-3), *ΔsbmA742::kan, λ^−^*, *rph-1,**Δ(rhaD-rhaB)568, hsdR514* (http://cgsc.biology.yale.edu/StrainRpt.php, accessed on 15 March 2022) were grown in Luria-Bertani (LB) at 37 °C under aerobic conditions with the addition, when required, of the appropriate antibiotics at the following concentrations—100 μg/mL for ampicillin and 50 μg/mL for kanamycin—which were purchased from Sigma-Aldrich (St. Louis, MO, USA) and diluted in LB medium.

*E. coli* CGSC strain 85447 lacking the SbmA importer expressing the ABC*tmf*A receptor/importer was prepared using plasmid pTAC-MAT-2 ABC*tmf*A importer/receptor as described earlier [7]. [3,4,5 ^3^H]Leucine 250 μCi, Spc Act 100–150 Ci/mmol was purchased from Perkin Elmer (Waltham, MA, USA). *Aea*TMOF-FITC (H-YDPAPPPPPPK(FITC)K-OH) was synthesized at the University of Colorado Anschutz School of Medicine protein core. *Aea*TMOF-FITC was purified by HPLC showing a single peak, and mass spectrometry analysis of the peak identified the 3 expected ions at 455.7, 607.3, and 910.3, showing an expected M_r_ 1019.56 as was described earlier [7]. Synthetic *Aea*TMOF (H-YDPAPPPPPP-OH) and a short Oncocin112(1–13) (H-VDKPPYLPRPRPP-OH), a 13 amino acids oncocin112 peptide, 6 amino acids shorter than the original oncocin112 of 19 amino acids, were synthesized and purified by HPLC [16]. After HPLC purifications of *Aea*TMOF, oncocin112 (1–13), and *Aea*TMOF-FITC, the TFA ions were exchanged with phosphate ions.

### 2.2. Cloning of Late Larval Ae. aegypti Trypsin

A synthetic *Ae. aegypti* late larval trypsin gene (accession number AY198134.1) (857 bp) with *Sgf*I and *Pme*I cloning sites at the 5′ and 3′ ends was synthesized by GeneScript (Piscataway, NJ, USA). The gene was cut with *Sgf*I and *Pme*I restriction enzymes (Promega Madison, WI, USA) following the manufacturer’s guidelines and cloned into a plasmid pFA25A ICE T7 flexi that was opened with *Sgf*I and *Pme*I following the manufacturer’s guidelines (Promega). The recombinant plasmid with the late larval trypsin gene was sequenced, confirming that a full-length *Ae. aegypti* larval late trypsin gene was cloned unidirectional at the 5′ to 3′ ends, replacing the Barnase gene on pFA25A ICE T7 flexi (Promega).

### 2.3. Inhibition of Luciferase Translation in S30 Bacterial Extract

To test the effect of *Aea*TMOF and Oncocin112 (1–13) on the translation of Luciferase by the bacterial 70 S ribosome, an *E. coli* S30 lysate extract system for circular DNA (Promega) containing a plasmid pBest*luc* DNA vector (4486 bp) (1 mg/mL) 0.3 μL, amino acid mixture without methionine (1 mM) 1.25 μL, amino acid mixture without Leucine (1 mM) 1.25 μL, S30 premix (including ribosomes) without amino acids 5 μL, S30 extract circular 3.5 μL, RNasin (8 U, 112 ng) 0.2 μL, different concentrations of *Aea*TMOF or Oncocin112 (1–13) (0.001 μM to 1000 μM), and nuclease-free water to a volume of 12.5 μL were incubated for 2 h at 37 °C. Control reactions did not contain pBest*luc* DNA. After incubation, 87.5 μL of GLO lysis buffer (Promega) containing beetle luciferin was added, mixed, and aliquots (5 μL) were removed, and their luminescence read in a GLOMAX multidetector system. Reactions were run in triplicates, and luminescence was corrected for control reactions that did not contain plasmid pBest*luc* DNA. Maximum luminescence (100%) was observed at 0.001 μM *Aea*TMOF or oncocin 112 (1–13) and was similar to incubations without the presence of the peptides. The decrease in luminescence (% ± SEM) in the presence of increasing log *Aea*TMOF or Oncocin112 (1–13) concentrations was then plotted using GrapPad Prism 3.

### 2.4. Inhibition of Larval Ae. aegypti Late Trypsin Translation in Insect Cell Extract

To test the effect of *Aea*TMOF on the translation of larval *Ae. aegypti* late trypsin, the TnT T7 insect cell extract protein expression system from *S. frugiperda* Sf21 cell line (Promega) was incubated with different concentrations of *Aea*TMOF in a reaction mixture containing TNT T7 ICE master mix of 10 μL, pFA25A ICE T7 flexi-late larval Trypsin (0.45 μg/mL) (see Section 2.2) 0.5 μL, [3,4,5 ^3^H]Leucine 0.5 μL, *Aea*TMOF (0.01 to 1000 μM), and nuclease-free water to a total volume of 12.5 μL. The incubation mixtures were incubated at 30 °C for 4 h. At the end of the incubation period, aliquots (10 μL) were removed and spotted on filter paper squares (1 × 1 cm) and washed in cold 10% TCA for 10 min, and twice in cold 5% TCA for 5 min each followed by 2 min with ethanol (100%). The paper squares were dried at 70 °C in an oven and tritium labeled trypsin counted in a liquid scintillation counter PerkinElmer Tri-Carb 2910 TR. Controls were run in the presence of empty plasmid pFA25A ICE T7 flexi that was not cloned with *Ae. aegypti* larval late trypsin and subtracted from the experimental results. Incubations were run in triplicates, and results are expressed as means ± SEM. The decrease in [^3^H]trypsin biosynthesis was then plotted against the increase in *Aea*TMOF concentrations using GraphPad Prism 3.

### 2.5. Western Blotting and Mass Spectra Analyses

#### 2.5.1. Western Blotting

To prove that the [^3^H]-labeled larval late trypsin synthesized by the TnT T7 insect cell lysate is indeed trypsin, the TnT T7 insect cell lysate was incubated at 30 °C for 4 h with Fluorotech Green tRNA_lys_ (0.4 μL) in a reaction mixture (12.5 μL) containing TnT T7 ICE master mix of 10 μL, pFA25A ICE T7 flexi-larval late Trypsin (0.64 μg/mL) of 0.5 μL, and nuclease-free water of 1.6 μL. The incubation was repeated 3 times. An incubation without pFA25A ICE T7 flexi-Trypsin was used as a control. After incubation, aliquots (5 μL) were removed from each incubation mixture and mixed with SDS sample buffer (4 μL) containing Tris-HCl, 2% SDS, 5% ME, and 20% glycerol with tracking dye. The incubation mixtures containing SDS-ME and glycerol were heated at 70 °C for 5 min and were loaded on a 10% Bis-Tris (1.0 mm) mini gel (ThermoFisher, Waltham, MA, USA), and electrophoresis was run at 110 V for 90 min [17]. A separate lane was run with broad molecular weight-stained standards (10 μL) (ThermoFisher). After SDS-PAGE, the 10% gel was removed, and proteins were transferred to 100% methanol wet PVDF membrane in Tris/CAPS buffer using Western blotting and a semidry transfer apparatus (BioRad, Hercules, CA, USA) at 15 V for 3 min following the manufacturer’s guidelines. The PVDF membrane was then scanned to detect fluorescence-emitting proteins using Sapphire Biomolecular Imager (Azure Biosystems, Dublin, CA, USA) at a wavelength of 468 nm.

#### 2.5.2. Mass Spectra Analysis

Four tubes containing four reactions mixtures (12.5 μL), TnT T7 ICE master mix (10 μL) and nuclease-free water (1.6 μL) were used. Tube 1 contained in addition BSA (40 μg; 0.5 μL), tube 2 contained only the master mix, tube 3 contained an empty plasmid pFA25A ICE T7 flexi (0.64 μg/mL; 0.5 μL), and tube 4 contained plasmid pFA25A ICE T7 flexi-late larval trypsin (0.64 μg/mL; 0.5 μL). The tubes were incubated at 30 °C for 4 h. After incubation, aliquots (5 μL) were removed from each incubation, put into sample buffer (4 μL) where the tubes were vortexed, heated at 70 °C for 5 min, and electrophoresed using SDS_PAGE as described above [17]. The gel was stained with Coomassie Brilliant Blue, and a protein band that ran at 28 kDa was cut from the destained gel, digested with trypsin, and analyzed by MS/MS [18].

### 2.6. Purification of E. coli Ribosomes

The purification of *E. coli* ribosomes is a modification of earlier published methods [19,20]. Briefly, *E. coli* (BL21) cells were grown in LB medium for 24 h to OD_600_ of 1.1, and the cells were centrifuged at 5000 rpm at 6 °C for 5 min. The supernatant was discarded and the pellet was washed with 40 mM HEPES-KOH pH7.4, 100 mM NaCl, and 2 mM Mg(OAc)_2_ buffer. The suspension was centrifuged at 6000 rpm for 5 min, the supernatant discarded, and the pellet was frozen at −80 °C and stored overnight. The frozen pellet was taken up in 10 mL of sonication buffer of 20 mM HEPES-KOH pH 7.5, 100 NH_4_OAc, 6 mM βME, 0.5 mM Na-EDTA, and 0.5% Tween 20 containing 100 units of RNase inhibitor, and the solution was sonicated at 40% power for 10 s, 50% power for 10 s, and 60% power for 20 s. During the sonication, the tube was immersed in an ice bath. The lysed cell solution (25 mL) in sonication buffer was centrifuged at 30,000 *g* in a Beckman ultracentrifuge using a 70Ti centrifuge rotor for 30 min at 4 °C. After centrifugation, the supernatant was collected, and the pellet discarded. High salt sucrose cushion (20 mM HEPES-KOH pH 7.5, 50 mM Mg(OAc)_2_, 100 mM NH_4_Cl, 6 mM βME, 0.5 mM Na-EDTA, and 1.1 M sucrose) solutions (25 mL each) containing 100 units of RNase inhibitor were added to the two centrifuge tubes. Each tube was carefully layered over the sucrose cushion, using a syringe with a needle, with the ribosomal solution (12.5 mL) obtained after the 30,000 *g* spin in 70Ti rotor. The tubes were centrifuged at 4 °C and 100,000 *g* (35,000 rpm) for 20 h. The clear pellets were each carefully layered with 0.5 mL of tight association buffer (20 mM HEPES-KOH pH 7.6, 6 mM Mg(OAc)_2_, 30 mM NH_4_OAc and 6 mM βME) to wash the top of the pellets, and the wash solutions were discarded. The pellets were then dissolved in tight association buffer (4.5 mL) and the tubes were shaken gently in bucket full of ice for 4 h. The solubilized pellets were centrifuged at 16,000 *g* in Eppendorf tubes for 10 min at 4 °C and aliquots (2 μL) were removed and diluted 1:100 in tight association buffer. The amount of the protein in the ribosome preparation and the ribosomal RNA was determined (2.28 mg/mL protein and 3.932 mg/mL ribosomal RNA). The ribosome preparation was then diluted with 10% glycerol and stored at −80 °C [19,20]. Aliquots (5 μL) of the purified ribosomes were then adsorbed unto carbon-coated 300 mesh of Cu/Rh grids and stained with 2% uranyl acetate. The grids were then observed by electron microscopy with 120 kV Talos L120C TEM at the CryoEM of the cancer center at the University of Colorado Anschutz Medical School.

### 2.7. Binding of AeaTMOF-FITC to E. coli Ribosome Assay and Kinetics

Purified *E. coli* ribosomes (1 μL, 9.4 μg) in tight association buffer (20 mM HEPES-KOH pH 7.6, 6 mM Mg(OAc)_2_, 30 mM NH_4_OAc, and 6 mM βME) were incubated in Eppendorf tubes containing 100 μL of tight association buffer and 5 μL of *Aea*TMOF-FITC (0–0.2 μM) (Specific activity 8333 Fluorescence Units/pmol) for 60 min at 30 °C with gentle shaking. After incubation, 12 μL of TCA (100%) was added, and the tubes were vortexed and incubated on ice for 10 min. The tubes were then centrifuged at 20,000 *g* for 5 min at 4 °C, and the supernatants were discarded. To the pellets, TCA (5%) in 100 μL of tight association buffer was added, the tubes were vortexed and centrifuged, and the supernatants discarded as described above. The washing by centrifugation with 5% TCA was repeated 3 times, and after the final centrifugation, the pellets were dissolved in 100 μL of PBS containing 0.4N NaOH, and the fluorescence of the dissolved pellets was read in a GloMax multidetector system using a blue filter (Excitation at 490 nm and Emission at 510–570 nm). All data were corrected for nonspecific fluorescence when incubations were run without *Aea*TMOF-FITC or with *Aea*TMOF-FITC and without ribosomes. The amount of *Aea*TMOF-FITC that bound the ribosome was determined from a linear calibration curve after plotting different concentrations of *Aea*TMOF-FITC (pmol) against fluorescence units. The results are expressed as *Aea*TMOF-FITC (pmol/μg ribosome). The kinetics of *Aea*TMOF-FITC binding to the purified bacterial ribosomes was followed at different concentrations of *Aea*TMOF-FITC using six independent experiments per concentration, the data was then fitted by nonlinear regression using GraphPad Prism 3 and the K_D_, B_max_ and the affinity constant K_assoc_ determined. The binding experiments were repeated two independent times.

### 2.8. Molecular Modeling and Docking

To dock *Aea*TMOF to the *Thermos thermophilus* 70S ribosome, we first docked Oncocin112 to the *T. thermophilus* ribosome using the coordinates in PBD (ID: 4Z8C) [15] (results not shown). The proline-rich *Aea*TMOF structure [21] was manually positioned in the adenine-binding site of the ribosome and superimposed to oncocin112, refined, and energy minimized using the YASARA structure [22]. To dock *Aea*TMOF to the *Drosophila melanogaster* and *E. coli* ribosomes, the peptide was built up with Chimera and minimized with 1000 cycles of steepest descent and 100 additional cycles of conjugate gradient, using the Amber force field [23]. The atomic coordinates of the Drosophila melanogaster 80S ribosome were taken from the PDB (4V6W) [24]. The atomic coordinates of the *E. coli* 70S ribosome (PDB code 6O9J) associated with the proline-rich antimicrobial peptide oncocin112 [15] were used for the docking of *Aea*TMOF to the *E. coli* and *Drosophila* ribosomes. Docking trials were performed with YASARA [22]. Molecular cartoons were drawn with Chimera [23] and ChimeraX [25].

### 2.9. Effect of AeaTMOF and Oncocin112 (1–13) in Vivo on E. coli Growth

We aimed to find out if *Aea*TMOF can also stop bacterial growth as, was shown for the proline-rich Oncocin112 that binds to the bacterial ribosome, stopping protein translation, bacterial growth, and causing bacterial death [14]. *E. coli* CGSC strain7636 (2.7 × 10^4^ cells) *SbmA*^+^, expressing the proline-rich peptides importer SbmA, were incubated in 96-well plate containing 100 μL/well of LB medium, 0.4 mM of IPTG, 0.1 mM of ampicillin, and different concentrations of *Aea*TMOF and Oncocin112 (1–13) (3 mM and 10 mM) in different wells. The 96-well plate was incubated at 37 °C for 18 h and bacterial growth was followed at 630 nm using Biotech ElX808 absorbance microplate reader. Control wells contained bacterial cells growing without *Aea*TMOF or oncocin112 (1–13). In a second experiment, *E. coli* mutant CGSC strain 8547 (2.7 × 10^4^ cells) *SbmA^-^* that do not express the Oncocin112 importer SbmA but do express the ABC *tmf*A importer [7] were incubated in 100 μL of LB medium with *Aea*TMOF and oncocin112 (1–13) (3 mM and 10 mM) as described above for 18 h following bacterial growth. After the incubations in the presence of Oncocin112 (1–13) or *Aea*TMOF of 10 mM and 3 mM, aliquots (0.5 μL) were removed from each well and spread on LB agar plates containing 0.1 mM of ampicillin, and the plates were incubated overnight at 37 °C and viable colonies were counted. Incubations were done in triplicates and the experiments were repeated twice. Results are expressed as means of three determinations ± SEM.

### 2.10. Statistical Analysis

Statistical analyses were determined using GraphPad Prism using linear and nonlinear regression. Results were considered significant when *p* < 0.05. Kinetic parameters K_D_ and B_max_ were determined from a nonlinear regression (*R^2^ >* 0.84) using GraphPad Prism. Each experimental point is a mean of 3 or 6 determinations ± SEM.

## 3. Results

### 3.1. Inhibition of Luciferase Biosynthesis in E. coli S30 Extract System

#### 3.1.1. Inhibition by *Aea*TMOF

Incubations of *E. coli* S30 extract synthesizing Beetle luciferase in the presence of an increasing concentration of *Aea*TMOF (0.001 to 750 μM) stopped the biosynthesis of luciferase, diminishing its activity on Beetle luciferin as measured in luminescence units. The activity of the in vitro synthesized luciferase on Beetle luciferin at very low concentration of *Aea*TMOF (0.001 μM) was similar to luciferase activity without *Aea*TMOF, and this luminescence was used to calculate the decrease in luminescence (%) after incubation with increasing concentrations of *Aea*TMOF (Figure 1A). Higher concentrations of *Aea*TMOF (higher than 0.001 μM) caused a linear decrease in the synthesis of luciferase and a decline in luminescence (%) reaching IC_50_ (inhibition concentration of 50%) at 1 μM (Figure 1A). Complete inhibition of luciferase biosynthesis was reached at *Aea*TMOF concentrations of 250–750 μM. *Aea*TMOF is not an inhibitor of luciferase. Incubations of the *E. coli* S30 extract system (Promega) in the presence of plasmid *pBESTluc* (Promega) for 4 h without *Aea*TMOF and then adding *Aea*TMOF (1 to 100 μM) did not affect the activity of luciferase that was synthesized in vitro as compared with controls without the addition of *Aea*TMOF (results not shown).

#### 3.1.2. Inhibition by Oncocin112 (1–13)

Incubations of *E. coli* S30 extract synthesizing Beetle luciferase in the presence of increasing concentrations of oncocin112 (1–13) (0.001 to 750 μM) also stopped the biosynthesis of luciferase and diminished its activity on Beetle luciferin measured in luminescence units as was shown in Section 3.1.1. The activity of the in vitro synthesized luciferase on Beetle luciferin at very low concentration of Oncocin112 (1–13) (0.001 and 0.01 μM) was the same and similar to luciferase activity without Oncocin112 (1–13), and this luminescence was used to calculate the decrease in luminescence (%) after incubation with increasing concentrations of Oncocin112 (1–13) (Figure 1B). Similar to the results obtained with *Aea*TMOF, Oncocin112 (1–13) also inhibits Beetle luciferase biosynthesis and not its activity (results not shown). The linear decline in the luminescence after incubation with Oncocin112 (1–13) reached an IC_50_ at 2 μM (Figure 1B), a twofold increase as compared with the *Aea*TMOF decapeptide. Oncocin112 (1–13) at 0.01 μM and *Aea*TMOF at 0.001 μM do not inhibit the biosynthesis of luciferase (Figure 1A,B).

### 3.2. Inhibition of Ae. aegypti Late Larval Trypsin Biosynthesis with AeaTMOF

To find out how *Aea*TMOF blocks trypsin biosynthesis in mosquitoes and other insects [4,5], increasing concentrations of *Aea*TMOF (0.01 to 750 μM) were incubated in the TnT T7 insect cell extract protein expression system from *S. frugiperda* in the presence of recombinant plasmid pFA25A ICE T7 flexi carrying *Ae. aegypti* late larval trypsin in the presence of [3,4,5 ^3^H]Leucine and [^3^H]trypsin biosynthesis. A control without *Aea*TMOF was also run showing that [^3^H]trypsin synthesized without *Aea*TMOF or in the presence of *Aea*TMOF (0.01 μM) was the same (results not shown). Increasing concentrations of *Aea*TMOF caused a rapid decrease in the synthesis of [^3^H]trypsin. A linear regression of the decrease in the synthesis was plotted against *Aea*TMOF (log μM) indicating that the IC_50_ of *Aea*TMOF is 1.8 μM (Figure 2). To prove that the [^3^ H]protein that was synthesized by the *S. frugiperda* 80S ribosomal lysate is trypsin, the protein expression system used above was first incubated with Fluorotech Green tRNA_lys_, and the synthesized fluorescent proteins were analyzed using Western blotting (Appendix A). A distinct fluorescence band (28.5 kDa) closer to the *Ae. aegypti* late larval trypsin 27.5 kDa, was detected in lanes b, c, and d after the insect protein expression system was incubated in the presence of Fluorotech Green tRNA_lys_ and was absent when the tRNA was not added to the incubation mixture (Appendix A lane a). Therefore, the fluorescence-emitting protein bands in lanes b, c, and d are probably the *Ae. aegypti* late larval trypsin (Appendix A). Incubating the insect cell extract protein expression system with (a) BSA, (b) only insect cell extract, or (c) pFA25A ICE T7 flexi (empty plasmid) and separating the proteins by SDS-PAGE (Appendix A, lanes 1, 2, and 3, respectively) produced faint bands at 28.5 kDa, but not as intense as the distinct protein band at 28.5 kDa that was observed when the insect cell extract protein expression system was incubated with pFA25A ICE T7 flexi carrying *Ae. aegypti* late larval trypsin (Appendix A, lane 4). MS/MS analysis of the 28.5 kDa band identified seven unique peptides with masses of 2633.31, 2060.1, 2373.13, 2072.94, 1187.61, 1209.57, and 775.358, and a 48% coverage of the protein (94 amino acids out of 250 amino acids) (Appendix A). The protein was identified by mass spectrometry as *Ae. aegypti* larval late trypsin (accession number AAO43403.1). The very faint and diffused stained bands in lanes 2 and 3 that ran at 28.5 kDa (Appendix A) were also analyzed by MS/MS but are not *Ae. aegypti* larval late trypsin.

### 3.3. Binding Kinetics of AeaTMOF to E. coli Ribosomes

To find out whether the ribosome is the ultimate target of *Aea*TMOF as was shown for oncocin112 [14,15], a purified ribosome preparation (2.28 mg/mL protein and 3.932 mg/mL ribosomal RNA) that was negatively stained showing no damage or dissociation after the purification was used (Figure 3). *Aea*TMOF-FITC bound the purified *E. coli* ribosome preparation showing concentration dependence, specific, and high-affinity binding (K_D_ = 23 ± 3.4 nM ± SEM and B_max_ = 0.553 ± 0.023 pmol/μg ribosome ± SEM) (Figure 4, n = 6). Using the specific binding (K_D_), an affinity constant was calculated (K_assoc_ = 4.3 × 10^7^ M^−1^). Nonspecific binding was subtracted from each point (see Section 2.7).

### 3.4. Three-Dimensional Modeling of AeaTMOF Binding to Bacterial Ribosome

#### 3.4.1. Docking to *T. thermophilus* Ribosome

For the initial molecular docking of *Aea*TMOF to the 70S ribosome of *T. thermophilus,* we used the X-ray crystallography of Onc112 binding to *T. thermophilus* 70S ribosome as a reference [14,15]. Docking of *Aea*TMOF to the 70S ribosome of *T. thermophilus* shows that the most favorable docking energy occurs when *Aea*TMOF interacted with the ribosome at the entrance to the peptide exit tunnel, just after the A-cleft of 50S RNA (Figure 5A). Several nucleotide bases (G2082, A2083, **U2517**, U2596, and **U2620**) create putative hydrogen bonds and hydrophobic stacking interactions with *Aea*TMOF (Figure 5B). Two of these nucleotide bases (**in bold**) participate in the binding of *Aea*TMOF to the ribosomal 50S RNA. Like Onc112 [14], the binding of *Aea*TMOF to the ribosomal 50S RNA probably interferes with the binding of the CCA end of the aminoacyl-tRNA, inhibiting the translation process by stopping the peptidyl transferase; however, more experiments will have to be done in the future to find out the exact mechanism.

#### 3.4.2. Docking to *D. melanogaster* and *E. coli* Ribosomes

Docking of *Aea*TMOF to *D. melanogaster* (60S) and *E. coli* (50S) ribosomes exhibited the most favorable energy when *Aea*TMOF occupied the peptide exit tunnel of both the *Drosophila* (60S) and *E. coli* (50S) ribosome large subunit (Figure 6A,C), blocking the entrance to the exit tunnel mainly by hydrophobic stacking interactions with the rings from purine and pyrimidine nucleotide bases of the *Drosophila* (28S) and *E.coli* (23S) r-RNA. A few hydrogen bonds between *Aea*TMOF and the nucleotide bases also participate in the interaction of *Aea*TMOF with the 28S and 23S r-RNAs (Figure 6B,D).

### 3.5. Inhibition of E.coli Growth with AeaTMOF and Oncocin112 (1–13)

Our results show that *Aea*TMOF binds to the *E. coli-*purified ribosome with high affinity (Figure 4), and that Onconcin112 (1–13) inhibits the biosynthesis of luciferase in vitro using the *E. coli* S30 lysate extract system for circular DNA (Promega) (Figure 1B). Oncocin112 was shown to kill *E. coli* by inhibiting bacterial protein translation at the 70S ribosome and using the SbmA transporter to get into the bacterial cytosol [13,26]. Our earlier results show that *Aea*TMOF enters bacterial cells that were engineered with the ABC-*tmf*A importer [7]. To find out if *Aea*TMOF-like oncocin 112 affects bacterial growth by entering the bacterial cell using SbmA importer, *E. coli* CGSC strain7636 cells expressing *SbmA*^+^ and *E. coli* mutant cells CGSC strain 8547 *SbmA^-^* expressing only ABC-*tmf*A importer were incubated with different concentrations of Oncocin112 (1–13) and *Aea*TMOF (3 and 10 mM). Both peptides significantly inhibited the growth of the *E. coli* cells, and only 4–10 colonies were still viable at the end of the incubation period from the initial 2.7 × 10^4^ cell, when compared with control cells that were not incubated with the peptides (*p <* 0.05) if the cells expressed a SbmA importer (Figure 7A). On the other hand, *E. coli* cells lacking a SbmA importer but expressing the ABC-*tmf*A importer were significantly (*p* < 0.05) inhibited by *Aea*TMOF (Figure 7B). Few colonies between 4 and 12 out of the starting 2.7 × 10^4^ colonies were still viable at the end of the incubation period. Control cells that were not incubated with the peptides grew at the same rate as cells that were incubated in the presence of Oncocin112 (1–13) (3 and 10 mM) (Figure 7B), indicating that *Aea*TMOF can use the mosquito ABC*tmf*A importer, as well as the bacterial SbmA importer. On the other hand, Oncocin112 (1–13) can only use the bacterial natural importer SbmA.

## 4. Discussion

*Aea*TMOF (YDPAPPPPPP) and *Neb*TMOF (NPTNLH) were shown to stop the translation of trypsin mRNA in the gut epithelial cells of *Ae. aegypti* and *Neobellieria bullata* [4,5]. *Aea*TMOF binds a gut-specific receptor ABC-*tm*fA transporter (accession number MK895491) that imports *Aea*TMOF from the hemolymph after it is secreted by the mosquito ovaries into the gut epithelial cells, stopping the translation of the trypsin transcript in the gut epithelial cells [1,2,7]. Several of these ABC transporters with 80–100% similarity with the *Aea*TMOF receptor are also found in several *Aedes, Anopheles,* and *Culex* species, and their trypsin biosynthesis is controlled by AeaTMOF [7,27]. However, the mechanism by which *Aea*TMOF stops the translation of the trypsin transcript in the midgut epithelial cells of *Ae. aegypti* is unknown. We observed that *Aea*TMOF belongs to a family of proline-rich peptides that were shown to act as antimicrobial peptides by inhibiting the translation of proteins by bacterial ribosomes like Oncocin112 [13,14,15]. Therefore, we performed in vitro transcript/translation experiments using the *E. coli* 30S lysate extract system (Promega) with increasing concentrations of *Aea*TMOF or Oncocin112 (1–13). The latter was shown by X-ray crystallography to occupy the ribosomal exit tunnel at its entrance, blocking tRNA movement, mRNA translation, and protein biosynthesis [14,15]. Our results (Figure 1A,B) show that increasing concentrations of *Aea*TMOF or Oncocin112 (1–13) in vitro inhibit the biosynthesis of luciferase (IC_50_ 1 μM and 2 μM, respectively). The peptides do not inhibit the activity of luciferase in the presence of *Aea*TMOF or Oncocin112 (1–13), indicating that the peptides act on the *E. coli* ribosome as was shown when Oncocin112 was incubated with *E. coli* ribosomes [14,15]. Although the full sequence of Oncocin112 (19 amino acids) was shown to inhibit in vitro protein biosynthesis [13], we decided to use a shorter Oncocin112 peptide, 13 amino acids long, because amino acids (1–13) occupy the entrance to the ribosomal exit tunnel [14,15] and a 13 amino acids peptide is closer in length to the decapeptide *Aea*TMOF [28]. *Aea*TMOF that was fed in vivo to *Ae. aegypti* larvae stopped larval trypsin biosynthesis and caused larval starvation and death when they were fed increasing amounts of *Aea*TMOF expressed by engineered *Chlorella desiccate*, *Saccharomyces cerevisiae,* and *Pichia pastoris* [29,30,31]. Our in vitro results also show that increasing concentrations of *Aea*TMOF incubated with *S. frugiperda* cell extract in the presence of plasmid pFA25A ICE T7 flexi engineered with *Ae. aegypti* late larval trypsin caused inhibition of the biosynthesis of [^3^H]trypsin (Figure 2). The IC_50_ (1.8 μM) that was determined is higher than the IC_50_ (1.0 μM) that was observed when *Aea*TMOF was incubated in vitro using *E.coli* S30 extract to synthesize luciferase (Figure 1A). Our results also show that *Aea*TMOF does not inhibit luciferase activity, and therefore does not interfere with the IC_50_ determinations. However, the two plasmids that were engineered to express luciferase and trypsin are different, and the transcription of luciferase and the late larval trypsin by the two plasmids is probably different, affecting the IC_50_. Indeed, different IC_50_ values from 0.150 μM to 0.250 μM were reported for oncocin112 when it was incubated in vivo with *E. coli* cells [13]. The synthesis of [^3^H]-labeled larval protein in vitro using a lysate extract of *S. frugiperda* ribosomes [32] in the presence of a plasmid carrying the larval trypsin gene does not prove that the late larval trypsin was synthesized. However, MS/MS analysis of the 28.5 kDa protein after the SDS-PAGE identified larval late trypsin only in incubations containing the plasmid pF25A ICE T7 flexi engineered with *Ae. aegypti* larval late trypsin. Several similar protein bands that were detected after staining of the SDS-PAGE gel in lanes (2–4) are from the *S. frugiperda* cell lysate that was used (Appendix A). In lane 1, BSA was added, showing a heavily stained BSA band (66 kDa). A second fluorescent band that was observed at 25 kDA (Appendix A) might be due to an apparent internal translation start [33]. However, our SDS-PAGE did not detect this protein band (Appendix A).

Our results suggest that *Aea*TMOF binds the *E. coli* ribosome similarly to the binding of the proline-rich peptide Oncocin112 that blocks the *T. thermophilus* ribosome exit tunnel, stopping protein biosynthesis [14,15]. For this reason, *Aea*TMOF ribosomal interaction studies were conducted using isolated intact *E. coli* ribosomes (Figure 3) that were incubated with increasing concentration of *Aea*TMOF-FITC. The binding of *Aea*TMOF to the ribosome shows high affinity (K_D_ = 23 ± 3.4 nM ± SEM, B_max_ = 0.553 ± 0.023 pmol/μg ribosome ± SEM and K_assoc_ = 4.3 × 10^7^ M^−1^) (Figure 4). A dissociation constant in the nanomolar range was also shown for other proline-rich peptides, oncocins, and apidaecins, which also bind the 70S bacterial ribosome [13]. Crystallization and X-ray diffraction of oncocin112 bound to the *T. thermophilus* 70S ribosome shows that the first 13 amino acids occupy the entrance to the exit tunnel of the 50S ribosome [14,15]. 3D modeling and docking of *Aea*TMOF to *T. thermophilus* shows that *Aea*TMOF-like Oncocin112 can also block the entrance to the exit tunnel of the 50S ribosome, interacting with several nucleotides by hydrogen bonding and hydrophobic stacking interactions, and thus can also stop protein synthesis and elongation by tRNA (Figure 5A,B). Molecular modeling and docking of *Aea*TMOF to the *D. melanogaster* (60S) and *E. coli* (50S) ribosomes shows that *Aea*TMOF also blocks the entrance to the exit tunnel of the 60S and 50S ribosomes by interacting with the *D. melanogaster* (28S) and *E. coli* (23S) r-RNA using hydrophobic and hydrogen bond interactions with the nucleotide bases (Figure 6A–D). The 3D models of *Aea*TMOF binding to bacterial and *Drosophila* ribosome (Figure 5 and Figure 6) suggest, for the first time, how *Aea*TMOF stops the translation of the trypsin transcript that was reported earlier [4,5].

The bacterial inner membrane protein SbmA is involved in the import of peptides, proline-rich peptides (including oncocin112), nucleic acids, antisense peptides, and several oligomers [34,35]. Incubation of *E. coli* CGSC strain7636 cells expressing *SbmA^+^* and *E.coli* CGSC strain 8547 cells *SbmA^-^*,which was engineered with *Aea*TMOF receptor/importer (ABC*tmf*A) [7] with oncocin112 (1–13) or *Aea*TMOF (5 mM and 10 mM), inhibited bacterial growth in cells that expressed SbmA^+^ (Figure 7A), as was reported earlier for the proline-rich peptide oncocin112 [35]. However, when the peptides were incubated with *E. coli* cells that only expressed ABC*tmf*A receptor/importer [7], oncocin112 (1–13) did not affect the cell growth, even at high concentrations of 10 mM, whereas *Aea*TMOF entered the bacterial cell using the ABC*tmf*A importer inhibiting protein synthesis and cell growth (Figure 7B). In mosquitoes, *Aea*TMOF was shown by cytoimmunochemical analysis [36] to target specific gut epithelial cell that express the ABC*tmf*A (TMOF-specific receptor/importer) [7]. These cells express trypsin, which is secreted into the gut lumen and digests the blood meal to free amino acids that are used for egg development [1]. We hypothesize that *Aea*TMOF in these trypsin-synthesizing cells binds to the ribosomes that synthesize trypsin and does not inhibit the biosynthesis of other proteins that are synthesized in other midgut epithelial cells that lack the *Aea*TMOF-specific receptor/importer [7,36]. Indeed, injection of *Aea*TMOF into female mosquitoes that took a blood meal only inhibits trypsin biosynthesis, and as a result the blood meal is not digested and egg development is inhibited, making the female mosquitoes sterile but not killing them [1,16,27,28]. This would suggest that *Aea*TMOF blocks all protein synthesis of gut epithelial cells that express the ABC*tmf*A receptor. Alternatively, *Aea*TMOF may block translation of only specific transcripts within these receptor-expressing cells. Further work is needed to determine if *Aea*TMOF blocks all protein synthesis in receptor-bearing gut epithelial cells or can selectively block synthesis of specific transcripts.

In conclusion, this report shows for the first time that *Aea*TMOF, after entering the bacterial cytosol using a SbmA importer or engineered mosquito ABC*tmf*A-receptor/importer expressed in the inner membrane of *E. coli* cells, binds with high affinity (K_D_ = 23 ± 3.4 nM ± SEM) to the bacterial 70S. 3D modeling indicates that *Aea*TMOF can also bind the *Drosophila* 80S ribosome and possibly the mosquito 80S ribosome in the gut epithelial cells. These observations explain, for the first time, how trypsin biosynthesis in blood-fed female mosquitoes is regulated. Our binding experiment results to bacterial and insect ribosomes do not show whether the translation of the trypsin transcript is immediately stopped, as was shown for Oncocin112, in which the binding to the bacterial ribosome at the A-site in the empty exit tunnel affects the elongation factor Tu and the ternary complexes blocking elongation [37]. More work is needed to study *Aea*TMOF binding to mosquito ribosomes, as was done using the *E. coli* ribosome as a model system.

## Figures and Tables

**Figure 1 biomolecules-12-00577-f001:**
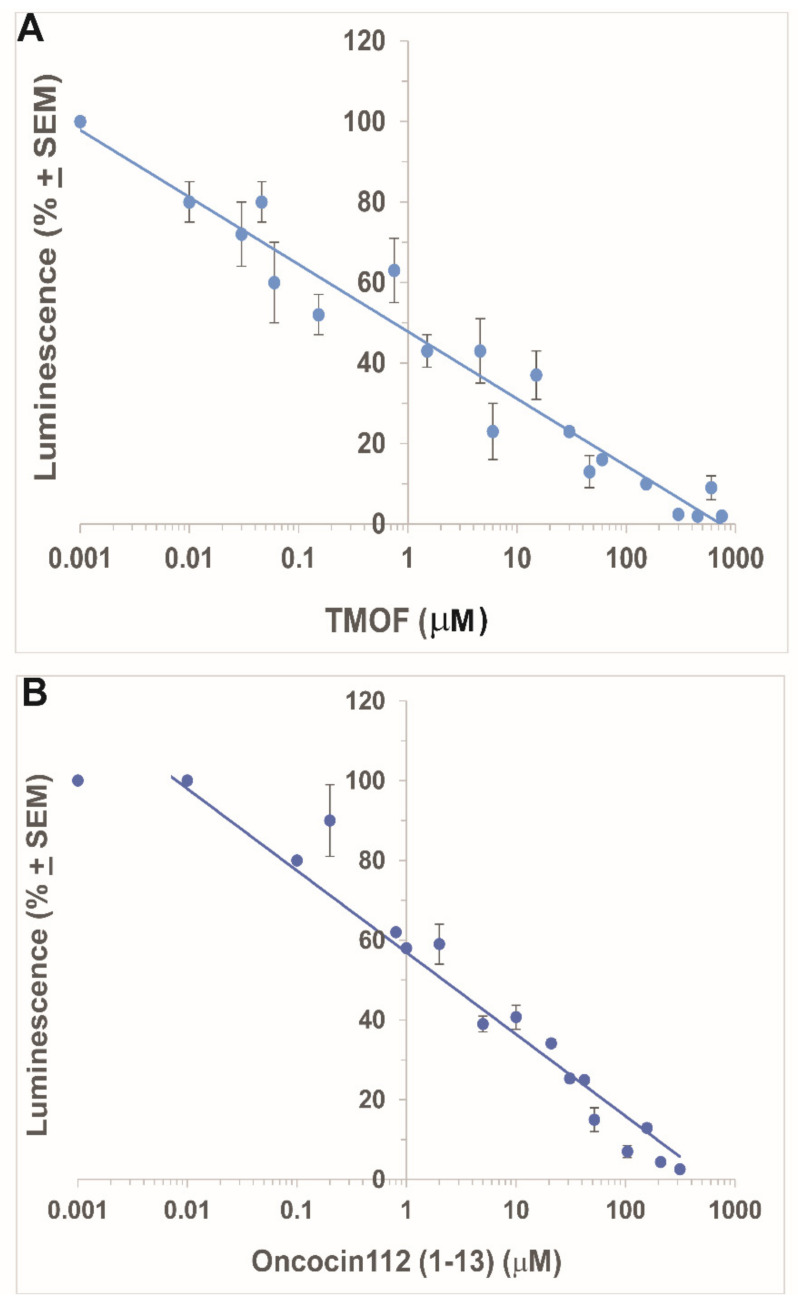
Inhibition of luciferase biosynthesis shown as reduction in luminescence (%) in *E. coli* S30 extract in the presence of increasing peptide concentrations (0.001-750 μM). (**A**) *Aea*TMOF showing IC_50_ of 1.0 μM, and (**B**) oncocin112 (1–13) showing an IC_50_ of 2.0 μM. Each point is the mean of three determinations + SEM (see Section 2.3 for details). Luminescence at 100% indicates no inhibition of luciferase biosynthesis.

**Figure 2 biomolecules-12-00577-f002:**
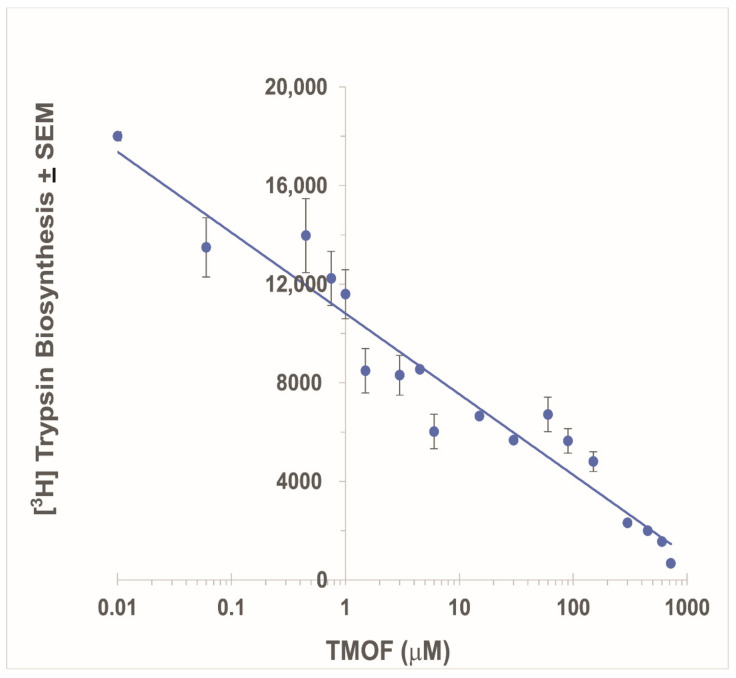
Inhibition of *Ae. aegypti* late larval [^3^H]trypsin biosynthesis by increasing concentrations (0.01–750 μM) of *Aea*TMOF in the *S. frugiperda* TnT T7 insect cell extract protein expression system, showing an IC_50_ of 1.8 μM. Each point is a mean of three determinations ± SEM (see Section 2.4 for details).

**Figure 3 biomolecules-12-00577-f003:**
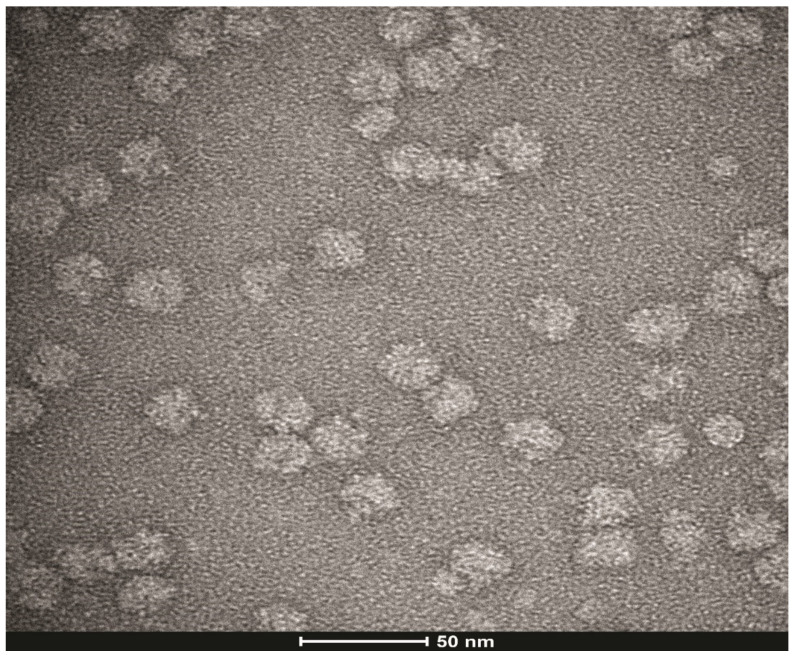
Purified *E. coli* ribosomes negatively stained with uranyl acetate on a carbon-coated grid and observed by electron microscopy with 120 kV Talos L120C TEM. The white bar below shows the relative size in nanometer. The 50S and 30S subunits can be observed in many of the ribosomes.

**Figure 4 biomolecules-12-00577-f004:**
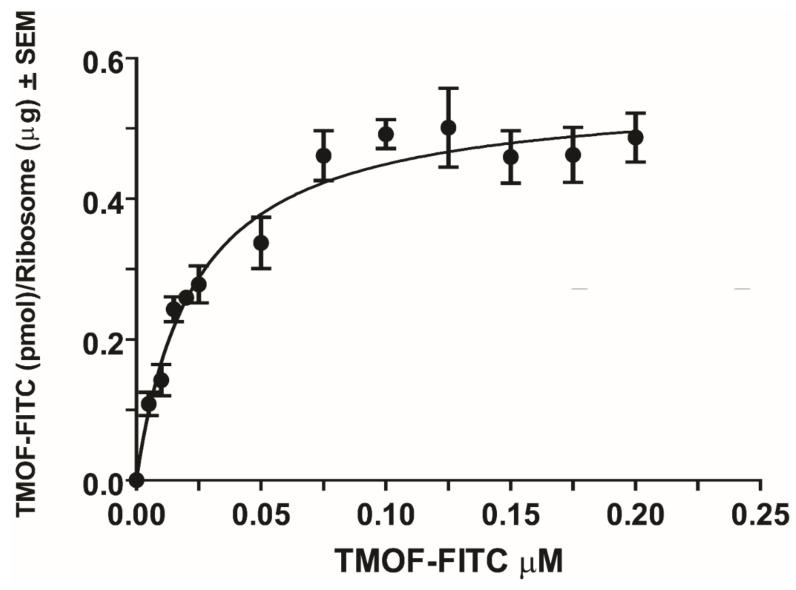
Specific binding of *Aea*TMOF to *E.coli-*purified ribosomes expressed as *Aea*TMOF-FITC (pmo/Ribosome (μg)) showing Michaelis Menten binding kinetics and using nonlinear regression (R^2^ = 0.84). Binding results were corrected for nonspecific binding by incubating ribosomes without *Aea*TMOF-FITC or with *Aea*TMOF-FITC and without ribosomes (see Section 2.7). Each point is an average of six determinations ± SEM. The data represent results of one experiment from two independent experiments with similar results. K_D_ = 23 ± 3.4 nM ± SEM, B_max_ = 0.553 ± 0.023 pmol/μg ribosome ± SEM, and K_assoc_ = 4.3 × 10^7^ M^−1^_._

**Figure 5 biomolecules-12-00577-f005:**
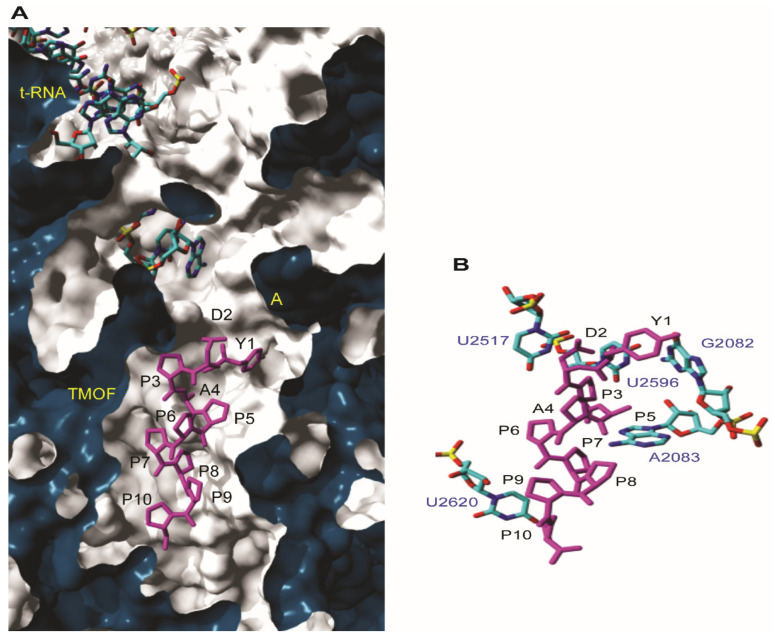
(**A**) Close-up cut view showing the docking of *Aea*TMOF (magenta stick) within the peptide exit tunnel of 23S RNA of *T. thermophilus* ribosome. The 5tRNA and the A-site of the 23S RNA are indicated. (**B**) Docking of *Aea*TMOF (magenta stick) into the peptide exit tunnel of 23S RNA of *T. thermophilus* showing the nucleotide bases (blue sticks) involved in the interaction with the peptide through hydrogen bonds (not shown) and stacking interactions.

**Figure 6 biomolecules-12-00577-f006:**
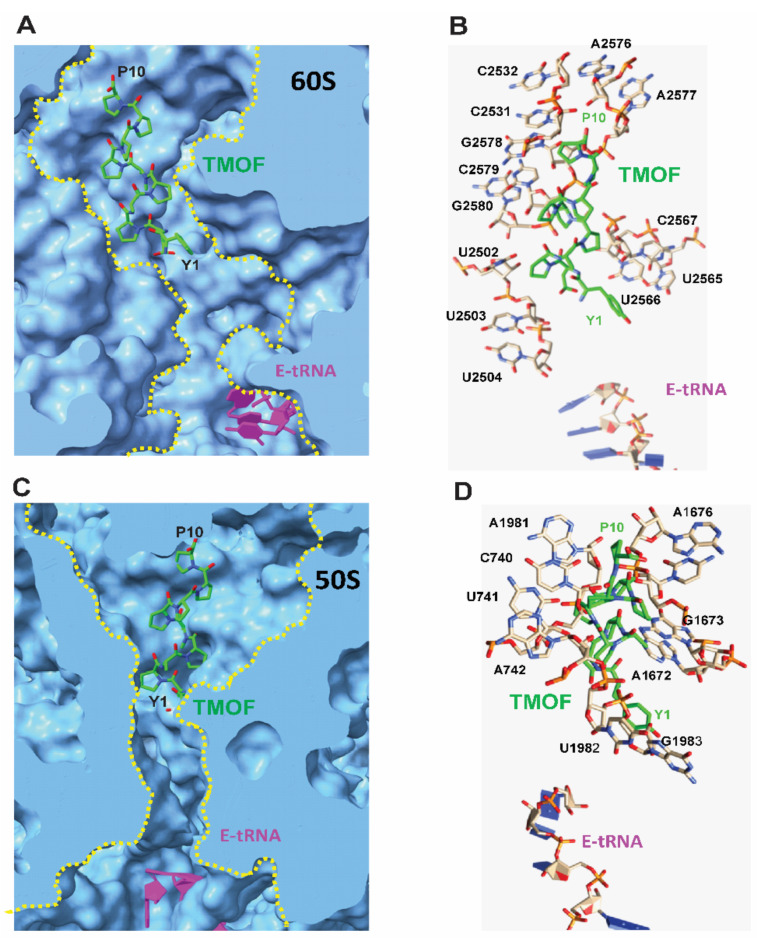
(**A**) Docking of *Aea*TMOF (colored green) to the Drosophila ribosome peptide exit tunnel (yellow dashed line) of the 60S large subunit (colored blue). The E-tRNA is colored magenta. (**B**) Cartoon showing the nucleotide bases of the 28S r-RNA from the 60S subunit that interacts with *Aea*TMOF via stacking interactions. The E-tRNA is also shown. (**C**) Docking of *AeaTMOF* (colored green) to the *E. coli* ribosome showing *Aea*TMOF (colored green) in the peptide exit tunnel (yellow dashed line) of the 50S large subunit (colored blue). The E-tRNA is colored magenta. (**D**) Cartoon showing the nucleotide bases of the 23S r-RNA from the 50S subunit that interacts with *Aea*TMOF via stacking interactions. The E-tRNA is also shown.

**Figure 7 biomolecules-12-00577-f007:**
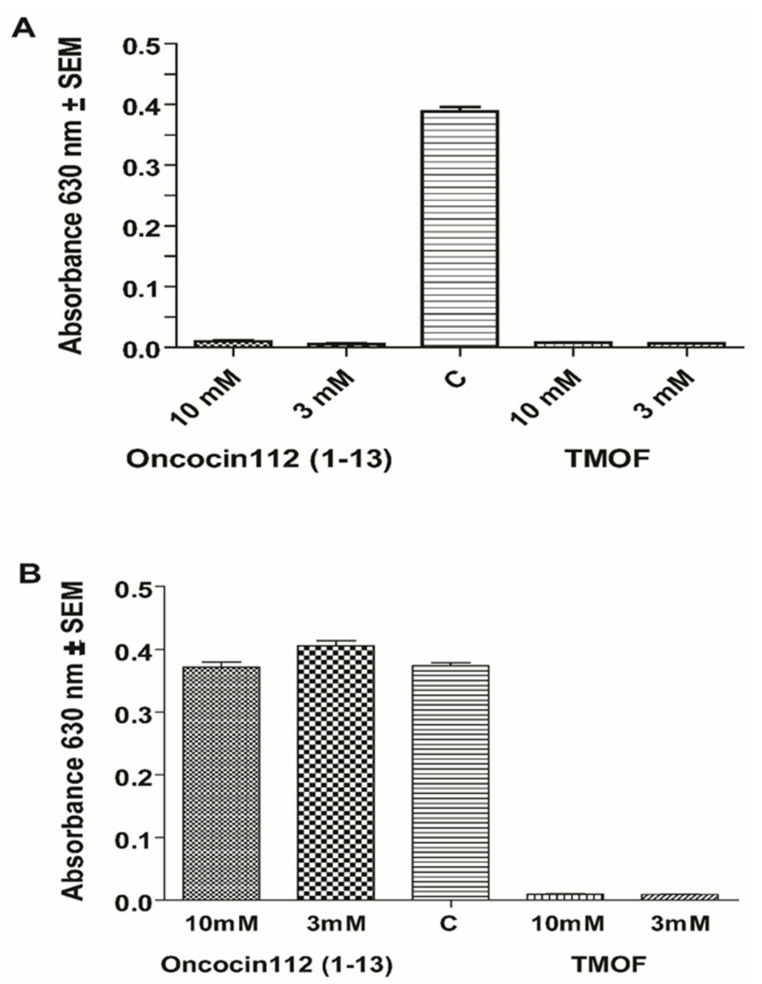
Incubation of *E. coli* cells with Oncocin112 (1–13) and *Aea*TMOF at concentrations of 3 and 10 mM (**A**) with *E. coli* cells expressing SbmA and (**B**) with *E. coli* mutant cells *SbmA*^-^ expressing the ABC-*tmf*A importer. Control cells incubated without oncocin112 (1–13) or *Aea*TMOF (C).

## Data Availability

Not applicable.

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
