# Peer review of "The Ribosome Is the Ultimate Receptor for Trypsin Modulating Oostatic Factor (TMOF)"

_biomolecules, 2022, doi:10.3390/biom12040577_

Round 1

Reviewer 1 Report

In this manuscript, authors investigate the trypsin modulating oostatic factor (TMOF). This manuscript includes potentially interesting findings suggesting that TMOF binds to the ribosome to inhibit translation and it is transported into gut epithelial cell through ABC-tmfA transporter. However, the manuscript is too sloppy. 

I am very confused by the inconsistency of “mM” and “µM” in the text and figures. “µM” appears in section 3.2 and Figs. 1 and 2, whereas “mM” appears in most other portions of text including figure legends. If KD for TMOF-FITC to E. coliribosome = 23 nM (Figure 4) is correct, IC50 for luciferase biosynthesis in E. coli S30 extract system = 1 mM (line 270) would be less reasonable than 1µM.  

In Fig. 4, “Bmax = --- pmol/mg ribosome” is described in the legend, but the title of the vertical axis is shown as “TMOF-FITC (pmol)/Ribosome (µg)”.

There are too many mistakes, e.g., spelling mistakes and grammatical mistakes. Before submission, the manuscript should be carefully checked by authors and then proofread by native English scientist.

Reviewer 2 Report

Borovsky et al report experiments to characterize the mechanism of action of TMOF. The synergy of biochemical and bacterial growth experiments as well as the expertise of the authors with the system are a strength of this manuscript. There are small inconsistencies and errors throughout the manuscript that need to be corrected. The authors also somewhat overstate their conclusions based on molecular docking calculations.

AeaTMOF should be spelled out in the first sentence of the abstract not later in line 19.

The naming of species is inconsistent throughout the introduction. By convention, the full name is given at the first occurrence and abbreviated afterwards. This is ignored on line 34, for example.

Why are lines 32, 33, 47, 48, 50-54 (and others later in the text) underlined? Could it be that a tracked word file was submitted and can it be ascertained that this was the final version for manuscript?

Line 305: The IC50 for the trypsin experiment is reported as 1.8 uM but the legend in Fig. 2 states it is 1.8 mM. Which is correct?

Figure 3 shows a simple confirmation that the purification of 70S ribosomes was successful. This figure does not provide any new results reported in this manuscript. It would therefore be more appropriate to show this in the supplementary materials.

Line 343: ‘the size in nano microns’ does not make sense.

Line 346: ‘Michaelis Menten binding kinetics’ there seems to be some confusion here about the difference between binding and enzymatic activity. This needs to be corrected.

L390: the sentence starting with ‘to find out if AeaTMOF…’ is very convoluted and therefore hard to read.  

Section 3.4:

The Onc112 crystal structure was used as a template for docking but the manuscript fails to show a comparison between Onc112 and the docked TMOF to illustrate any similarities or differences between the template and the docking results. This needs to be shown at least in the supplementary materials. In the same section, the sequence similarities between the two peptides need to be outlined to document that Onc112 is actually a viable template.

L 368-370: Based on a docking result, the authors conclude that TMOF binding interferes with the binding of the aminoacyl-tRNA CCA end. The experimental data presented do not support such a definitive statement. The docking results are suggestive without doubt but more data would be needed for a definitive conclusion. The same comment applies to the discussion in L474-480.

Typos

Line 68 ocncocin

  1. 92 oncoscin

L 365: several …bases…creates…

L 425: E. coli 30S extract

Reviewer 3 Report

In this article, the authors continue to investigate the role and mechanism of AeaTMOF. The authors showed that this polypeptide is able to block translation by binding to ribosomes. However, I think that ribosome, in this case, isn't the receptor, and the peptide isn't the ligand. This peptide is an unspecific translation inhibitor, and the expression of the ABC-tmfA transporter gives the specificity.

In my opinion, these results are sufficient, and the experiments are well designed. However, I would like to notify you about the weak presentation of data. Additionally, I have found a few mistakes and false statements:

Fig.2- The inhibition showed not in %.

Fig.3 - The white bar below shows the relative size not in nano microns, but nanometers.

In Supplementary Fig.2, PAAG staining by Coomasie does not detect any difference between lines 2, 3, and 4. This difference shouldn't be mentioned in the text (as well as the non-detectable line at 25kDa).

Reviewer 4 Report

The proline-rich peptide TMOF is isolated from Ae. aegypti  ovaries and control the synthesis of trypsin-like enzymes in the midgut of several insect species. In the current manuscript Borovsky, Rougé, and Shatters show that TMOF bind to E. coli ribosomes and inhibits the translation by E. coli and

Spodoptera frugiperda ribosomes in vitro. Furthermore, they model TMOF in the ribosome tunnel based on previously documented structures of ribosomes with Oncocin, another proline-rich peptide. Based on this, they propose that the ribosome is the ultimate targeted receptor of AeaTMOF. Although this would be an important conclusion, it is too little evidence to justify publication of the current manuscript. The authors might for example determine in the TMOC inhibits the transition from initiation to elongation in vitro.

Round 2

Reviewer 1 Report

Line 182-214

Mg(OAC)2 should be Mg(OAc)2.

Lines 232-236, 272-275, 355-359, Figure 4

How was the SEM for KD or Bmax determined? Is it from 6 independent KD or Bmax values or from two KD or Bmax values, each determined from 6 average data?

 “Each experimental point is a mean of 3 to 6 determinations + SEM” is described in line 272-275, whereas “each point is an average of six determinations + SEM” is described in Fig. 4 legend.

Why doesn't SEM appear in Kassoc? I am not sure if Kassoc is requisite.

Line 240

What is “adenine-binding site”? It does not seem popular.

Line 287

“mM” should be “micro M”.

Figure 6

What is E-tRNA? Is it E-site tRNA? If so, it should be deacylated tRNA rather than peptidyl-tRNA. However, “peptidyl-E-tRNA” appears in the legend. This term would not be popular. If it is a kind of peptidyl-tRNA, it should be in the P-site (before peptidyl-transfer or after translocation) or A-site (after peptidyl-transfer but before translocation). Various types of ribosomes have been used for structural study. What type of ribosome is used for docking in this study?

What residues of tRNA are shown in the figure? Are they 3’CCA terminal? Is it in the A site or P-site? If it is a peptidyl-tRNA, where is the peptide moiety?

Figures 5 and 6

Does docking model explain how AeaTMOF specifically binds to the ribosome (more tightly than other peptides)? Is it mainly due to stacking interaction of proline residues with base or CH-pi interaction?

Line 308-310

Please check the grammar of this sentence.

Lines 489-493

It is hard to understand what the authors want to say. I recommend rewriting. “(Figure 2A supplementary materials)” should be moved to the last of the first sentence.

Lines 526-528

This sentence sounds strange. The present manuscript only shows that AeaTMOF binds the ribosome and inhibits translation, but there is no binding site data. Figures 5 and 6 are entirely based on an earlier X-ray crystallographic study of another proline-rich peptide in complex with the ribosome. In addition, there is no data suggesting a mechanistic insight into the action of AeaTMOF in the ribosome.

Lines 552-553

The present study shows that AeaTMOF inhibits not only trypsin translation but also luciferase translation. This could rule out the possibility that AeaTMOF inhibits trypsin translation specifically rather than general translation in gut cell of Aedes.

Lines 562-566

Please check the grammar of this sentence.

Figure 1 supplementary material

Please describe the difference between lanes b, c and d.

The notation “AeaTMOF” is used in some places and “TMOF” is used in other places. I wonder if they are the same. If they are the same, please unify.

Author Response

Dear reviewer.

Thanks for all you constructive suggestions and questions.  I am listing all the corrections and answers to all your questions.  Your help and suggestions to make the manuscript better are appreciated.

Dov Borovksy, Professor

Answers to reviewer 1

Line 182-214

Mg(OAC)2 should be Mg(OAc)2. 

 Answer: See our corrections in line 197

Lines 232-236, 272-275, 355-359, Figure 4

How was the SEM for KD or Bmax determined? Is it from 6 independent KD or Bmax values or from two KD or Bmax values, each determined from 6 average data?

Answer:  KD and Bmax were determined from 6 independent experimental data points average of 6 determinations + SEM .  I made the correction to clarify this see line 240.  Although we also got similar results when the experiment was repeated a second time but the KD and Bmax shown in Figure 4 are not an average of these 2 experiments.  I believe that the legend in Figure 4 is clear about it see line 378-379.

 “Each experimental point is a mean of 3 to 6 determinations + SEM” is described in line 272-275, whereas “each point is an average of six determinations + SEM” is described in Fig. 4 legend.

Answer:  This sentence is taken from section 2.10 that sums all the experiments that were done in this manuscripts.  For example, some experiments were done using 3 determinations + SEM see Figures 1 and 2, whereas Figure 4 points were determined as 6 independent determination + SEM.  We have also repeated the experiment in Figure 4 one more time showing the same results.  We show the results of only one experiment in Figure 4 and the KD and Bmax are for this experiment, they are not averages of 2 KD  or 2 Bmax.

Why doesn't SEM appear in Kassoc? I am not sure if Kassoc is requisite.

Answer:  By definition Kassoc is the 1/M of the determined KD and thus SEM is not needed.  I believe that the Kassoc is of TMOF binding to the ribosome is a useful determination so the association of TMOF to the ribosome can be compared with TMOF association with its receptor (See Reference 7). 

Line 240

What is “adenine-binding site”? It does not seem popular.

Answer:  In prokaryotes the ribosomal protein S1 binds to adenine sequences upstream of the ribosomal binding site and this site is referred in our manuscript in line 247 when the docking was done.

Line 287

“mM” should be “micro M”.

Answer:  The mM was corrected into microM see line 295 in the revised manuscript

Figure 6

What is E-tRNA? Is it E-site tRNA? If so, it should be deacylated tRNA rather than peptidyl-tRNA. However, “peptidyl-E-tRNA” appears in the legend. This term would not be popular. If it is a kind of peptidyl-tRNA, it should be in the P-site (before peptidyl-transfer or after translocation) or A-site (after peptidyl-transfer but before translocation). Various types of ribosomes have been used for structural study. What type of ribosome is used for docking in this study?

Answer:  The E-tRNA is the at the E site of the ribosome and is deacylated and the peptidyl-E-tRNA has been changed to E-tRNA as the reviewer suggested.  The legend in Figure 6 A show that the docking of TMOF was done on Drosophila ribosome whereas in Figure 6C it was done using E. coli ribosome  see lines 410 and 413.

What residues of tRNA are shown in the figure? Are they 3’CCA terminal? Is it in the A site or P-site? If it is a peptidyl-tRNA, where is the peptide moiety?

Answer:  The tRNA is the 3’CCA terminal at the E site.  There is no peptide moiety on the tRNA at the E-site.

Figures 5 and 6

Does docking model explain how AeaTMOF specifically binds to the ribosome (more tightly than other peptides)? Is it mainly due to stacking interaction of proline residues with base or CH-pi interaction?

Answer:  The docking model shows that TMOF and Oncocin block the entrance to the exit tunnel preventing the binding of formyl-Met-tRNA to the A site and thus prevents elongation and protein biosynthesis.    Reference 38 and see also our discussion lines 585-589 explain how the inhibition takes place.  The interaction of TMOF is mainly due to hydrogen bonding and stacking interactions and interactions of the prolines with various bases (see lines 382-385, 393-398, 405-409 and 411-420)

Line 308-310

Please check the grammar of this sentence.                                                                            

 Answer: The sentence was rewritten see lines 323 -324 in the revised manuscript

Lines 489-493

It is hard to understand what the authors want to say. I recommend rewriting. “(Figure 2A supplementary materials)” should be moved to the last of the first sentence.

Answer:  The sentence that the reviewer suggested to change was re written see lines 507-510 in the revised manuscript.

Lines 526-528

This sentence sounds strange. The present manuscript only shows that AeaTMOF binds the ribosome and inhibits translation, but there is no binding site data. Figures 5 and 6 are entirely based on an earlier X-ray crystallographic study of another proline-rich peptide in complex with the ribosome. In addition, there is no data suggesting a mechanistic insight into the action of AeaTMOF in the ribosome.

Answer:

Only Figure 5 is based on earlier results with Oncocin 112 whereas Figure 6 docking results shows for the first time that TMOF can also bind to the entrance of the exit tunnel of E. coli and Drosophila in the same fashion as was shown for the proline rich peptide Oncocin 112 binding to Thermos thermophilus in Figure 5. Although we do not have yet CryoEM results showing binding TMOF to Drosophila , E. coli and Aedes aegypti ribosomes this is something that we plan for future studies.  The inhibition studies strongly suggest that TMOF binds to the ribosome and it very likely that the mechanism is similar to the action of Oncocin, and the discussion is the appropriate place to suggest it.  I have changed the wording in the sentence see line 556-557 starting “  The 3D models ……………….(Figures 5, 6) suggests……………..I changed explains with suggests.

Lines 552-553

The present study shows that AeaTMOF inhibits not only trypsin translation but also luciferase translation. This could rule out the possibility that AeaTMOF inhibits trypsin translation specifically rather than general translation in gut cell of Aedes.

Answer:  TMOF can block any protein biosynthesis by blocking to the ribosome in an in vitro  system that uses ribosome to translate mRNA.  HOWEVER,  in vivo TMOF can only enter an insect cells e.g., mosquito gut epithelial cells if they have a TMOF receptor,  these cells are  found  only in certain gut cells that synthesize trypsin during blood digestion 10-48 h after the blood meal.  The inhibition of the translation of luciferase was done in vitro not in vivo and this explains our results.  If luciferase was translated in a cell that lacks TMOF receptor importer TMOF would not enter that cell and inhibit its translation.  Thus, the discussion section that explains this phenomenon is correct see lines 572-591.

Lines 562-566

Please check the grammar of this sentence.

Answer:  The sentence was edited to make it grammatically correct.

Figure 1 supplementary material

Please describe the difference between lanes b, c and d.

 Answer:  Lanes b, c and d are the same .  They are three repeats of the same incubation that was described in the Figure legend.  This was done to show that the results our consistent.

The notation “AeaTMOF” is used in some places and “TMOF” is used in other places. I wonder if they are the same. If they are the same, please unify.

Answer:  The notation AeaTMOF has now replaced TMOF throughout the manuscript.

Reviewer 4 Report

I am sorry that the authors were disappointed about my lack of comments on the results in their manuscript. I can see that my review should have included statements to the effect that the experiments are well-conceived and executed, and that the results are precise and clear. However, my main concern was and still is, that the manuscript would be much stronger and more interesting if the mechanism for TMOF inhibition of translation were addressed. Furthermore, the in vitro experiments reported in the manuscript suggest that the authors are quite experienced in manipulating in vitro translation systems and presumably easily can address my concerns.

But it is always a judgment when a story is developed sufficiently to warrant publication. Since the authors insist, I suggest that the manuscript is published as-is.

Author Response

 Reviewer 4 approved the revised manuscript

We would like to thank this reviewer for his 

costructive suggestions.